# Impact of the Non-Uniform Catalyst Particle Size on Product Selectivities in Consecutive Reactions

**Juan Rafael García [1], Claudia María Bidabehere [2] and Ulises Sedran [1,\*]**

[1]  Instituto de Investigaciones en Catálisis y Petroquímica "Ing. José Miguel Parera" INCAPE (UNL–CONICET) Colectora Ruta Nac. Nº 168 Km 0—Paraje El Pozo, Santa Fe 3000, Argentina

[2]  Instituto de Investigaciones en Ciencia y Tecnología de Materiales INTEMA (FI, UNMDP–CONICET)—Juan B. Justo 4302, Mar del Plata 7600, Argentina

**\***  Correspondence: usedran@fiq.unl.edu.ar; Tel.: +54-(342)-451-1370 (ext. 6102)

**Abstract:** The analysis of consecutive reactions $A \rightarrow B \rightarrow C$ in porous catalyst particles, where the simultaneous processes of diffusion and chemical reactions take place and both reactant and products are subjected to diffusion limitations, was performed for catalyst particles with non-uniform sizes, a fact that has not been considered so far. The system comprises first-order consecutive irreversible reactions that proceed on spherical catalyst particles with a log-normal volume particle size distribution (PSD), which is typical in many catalytic applications. Regardless of the prevailing diffusion regime (chemical control, transition situation or intraparticle diffusion control), the yield of the intermediate product ($B$) reaches a maximum value as a function of the conversion of reactant ($A$), then decreases as a consequence of the prevalence of the secondary reaction that converts it into the secondary product ($C$). If intraparticle diffusion resistances affect the reactant species, given the relationship between the kinetic constants and the mean particle size, the selectivity to the intermediate product is negatively affected by the dispersion in PSD. The larger the dispersion in PSD, the stronger the negative impact.

**Keywords:** particle size distribution; consecutive reactions; selectivity; intermediate product

## 1. Introduction

Important examples of series reactions can be found in the chemical process industry, where maximizing the yield of the intermediate product is intended, as it is more important than the other products [1,2]. This concept includes multicomponent systems, where the reactant(s) and intermediate and final product(s) are indeed very complex mixtures and lumping methodologies are used [3]. Among them, for example, it is clear that intermediate fuels or olefins are the product of interest in the processes of hydrocracking [4] or catalytic cracking of hydrocarbons [5] and the conversion of methanol to hydrocarbons [6]. The oxidation of *n*-butane to maleic anhydride in the petrochemical industry is a remarkable illustration of consecutive reactions [1], with the primary product, maleic anhydride, being the desired one instead of the final oxidation products. The partial oxidation of methane, leading to the intermediate product of methanol, more extended oxidation products, such as formaldehyde and formic acid, or final $CO_2$ and $H_2O$, represents another example [2]. The selectivity issue in consecutive (series) reactions $A \rightarrow B \rightarrow C$, catalyzed by porous solids, has focused on the intermediate product $B$, as it may be converted into less appreciated products. The consequences of diffusion limitations on selectivity problems were extensively studied by, for example, Wheeler [7], Weisz et al. [8,9], van de Vusse [10], Vayenas and Pavlou [11], Sutradhar et al. [12], Szczygieł [13], García et al. [5,14–16], and Kang et al. [2].

It is well known that the intraparticle diffusion resistances are an intrinsic property of porous catalyst particles, which, under given reaction conditions, depend on pore size,

shape, tortuosity and constrictions, molecular size and configuration and diffusion length, with all these factors affecting the magnitude of a diffusivity parameter. Models, both fundamental and applied, were developed, which propose approaches with different complexity [17]. These issues severely affect the important concept of the catalytic effectiveness factor, which shows the working efficiency of active sites in a porous solid catalyst and simplifies chemical reactor analysis and design [18]. The effectiveness factor is expressed by the well-known Thiele modulus $\phi$, which includes diffusion and kinetics parameters [18–22], with the size of catalyst particles being a major issue in determining its magnitude, as it represents the diffusion length.

However, usually, the calculations of the effectiveness factors are founded upon the assumption that all the catalyst particles are the same size, a comfortable tool, provided the range in the sizes is narrow [23,24]. On the contrary, if the evidence of non-uniform sizes cannot be ignored, the analysis of the effectiveness factor is more difficult and criteria to quantify intraparticle diffusion limitations may become ambiguous [24]. For example, the well-known parameter by Weisz and Prater [8], which is the relationship between the observed reaction rate and a characteristic diffusion rate [18], is used to determine diffusion restrictions [19] and intrinsic kinetic constants when intraparticle diffusion limitations exist, if the catalyst particles are uniform in size [20,22,25–27]. If a particle size distribution exists, a characteristic dimension must be defined to represent all the particles, so as to avoid errors in the estimation of kinetic parameters [24]. Moreover, it is obvious that particles with different sizes will show different catalytic effectiveness [23,24].

Different methods exist to determine the particle size distribution (PSD) of catalyst particles, such as electron microscopy [28], laser diffraction analysis [29], light scattering [30], elutriation [31,32], or sieving [33]. It is common practice to assume log-normal volume distributions of the particle sizes [29,32,34], meaning that the logarithm of the particle diameter has a Gaussian distribution [35].

It is the objective of this work to study the consequences derived from the occurrence of a catalyst particle size distribution in a catalytic bed on the selectivity to intermediate products in consecutive (series) chemical reactions of the type *A→B→C*, under diffusion limitations. The influences of PSD on the yield curves of product *B*, according to the mean Thiele modulus and the relative magnitude of the kinetic constants, are analyzed and compared against the case of uniform catalyst particle size.

## 2. Theoretical Background: Uniform Catalyst Particle Size

### 2.1. System Description

The system under analysis includes the following consecutive reactions:

$$A \xrightarrow{\ k_1\ } B \xrightarrow{\ k_2\ } C \tag{1}$$

*B* and *C* are the primary and secondary products, respectively.

The reactions, which are assumed to be irreversible and are first-order reactions, take place over spherical porous catalytic particles in a reactor where only the reactant *A* is fed with inlet concentration $C_A^\circ$. The system is assumed to be isothermal and the adsorption equilibrium is linear; thus, $k_1$ and $k_2$ are overall kinetic constants, including Henry's law adsorption equilibrium constants. The catalyst deactivation is disregarded and the resistance to mass transfer in the film around the particles is negligible. Furthermore, the concentration profiles in the catalyst particles are assumed to be steady. Diffusion of reactant species in the pore system of the catalyst particles obeys Fick's law and the diffusion coefficients for reactant *A* and primary product *B* in the catalyst pores are assumed to be the same $\left( D_A = D_B = D \right)$.

### 2.2. Mass Balances

The mass balance for reactant *A* within a spherical catalyst particle with radius *R* is

$$D\left[\frac{1}{r^2}\frac{d}{dr}\left(r^2\frac{dC_{A(r)}}{dr}\right)\right]=k_1 C_{A(r)} \qquad (0 < r < R) \qquad (2)$$

$$\frac{dC_A}{dr}=0 \qquad (r=0) \qquad (3)$$

$$C_A = C_{Af} \qquad (r=R) \qquad (4)$$

and for the intermediate product B, it is

$$D\left[\frac{1}{r^2}\frac{d}{dr}\left(r^2\frac{dC_{B(r)}}{dr}\right)\right]=-k_1 C_{A(r)}+k_2 C_{B(r)} \qquad (0 < r < R) \qquad (5)$$

$$\frac{dC_B}{dr}=0 \qquad (r=0) \qquad (6)$$

$$C_B = C_{Bf} \qquad (r=R) \qquad (7)$$

The reduction in the number of parameters that control these systems and their simpler analysis can be achieved by means of the dimensional analysis approach [36]. By defining dimensionless variables in Equations (A1)–(A3) (see Appendix A), the following dimensionless parameters characterize the reacting system:

$$\phi_{(R)} = L_{(R)}\sqrt{\frac{k_1}{D}} = \frac{V_{p(R)}}{A_{p(R)}}\sqrt{\frac{k_1}{D}} \qquad (8)$$

and

$$m = \frac{k_2}{k_1} \qquad (9)$$

where $\phi_{(R)}$ is the generalized Thiele modulus for the primary reaction ($A\xrightarrow{k_1}B$) [37] and $m$ is the relationship between the intrinsic kinetic constants for the secondary reaction ($k_2$) and the primary reaction ($k_1$) [9].

*2.3. Conversion, Yield, and Selectivity*

Conversion can be calculated from the observed concentration of reactant A in the fluid phase ($C_{Af}$) as

$$X_A = \frac{C_A^\circ - C_{Af}}{C_A^\circ} \qquad (10)$$

or from the corresponding flux-based equation.

The relationship between the moles of product B observed in the fluid phase and the moles of reactant A fed into the reactor shows the yield of that product as

$$Y_B = \frac{C_{Bf}}{C_A^\circ} \qquad (11)$$

Selectivity to product B can be expressed as the relationship between its overall rate of formation and the rate of consumption of reactant A [19] and, after the mass balances in the fluid phase of the reactor, it is easy to show that

$$S_B = \frac{\int_{V_p}\left(k_1 C_{A(r)} - k_2 C_{B(r)}\right)dV}{\int_{V_p}\left(k_1 C_{A(r)}\right)dV} = \frac{-dC_{Bf}}{dC_{Af}} \quad (12)$$

Weisz and Swegler [9] used Equation (12) to analyze selectivity in this system, which can be written as a function of $X_A$ (Equation (10)) and $Y_B$ (Equation (11)) as

$$S_{B(R)} = \frac{dY_B}{dX_A} = \left\{\left(\frac{1}{m-1}\right) - \left(\frac{Y_B}{1-X_A}\right)\right\}\left(\frac{\phi_{(R)}\sqrt{m}\coth\left(\phi_{(R)}\sqrt{m}\right)-1}{\phi_{(R)}\coth\left(\phi_{(R)}\right)-1}\right) - \left(\frac{1}{m-1}\right) \quad (13)$$

and solved considering that the initial condition is

$$Y_B(X_A = 0) = 0 \quad (14)$$

Then, for a given pair of parameters $\phi_{(R)}$ (Thiele modulus for the primary reaction) and $m$ (relationship between kinetic constants, $k_2/k_1$), the yield of product $B$ can be calculated as a function of the conversion of reactant $A$ as

$$Y_B = \int_0^{X_A} S_B\, dX_A \quad (15)$$

or, by introducing Equation (13), as

$$Y_B = \int_0^{X_A}\left(\left\{\left(\frac{1}{m-1}\right) - \left(\frac{Y_B}{1-X_A}\right)\right\}\left(\frac{\phi_{(R)}\sqrt{m}\coth\left(\phi_{(R)}\sqrt{m}\right)-1}{\phi_{(R)}\coth\left(\phi_{(R)}\right)-1}\right) - \left(\frac{1}{m-1}\right)\right)dX_A \quad (16)$$

Figure 1 shows the yield of primary product $B$, as calculated from Equation (16), as a function of the conversion of reactant $A$, for different values of the Thiele modulus for the primary reaction ($\phi$) and relationships between kinetic constants ($m$). A similar behavior to this has been recently reported by Valecillos et al. [6] in the conversion of methanol to hydrocarbons. The authors observed that the yield of light olefins reached a maximum value of about 47% at 91% conversion of methanol, then decreased at higher conversions. This is certainly expected for the methanol to hydrocarbons reaction, as light olefins are intermediates in the kinetic scheme [38,39] and the continuous increases in the yields of light paraffins, heavy aliphatics and aromatics at high conversions indicate that these are the final products [6].

It can be observed in Figure 1 that under extreme conditions, that is, pure chemical control or net diffusion control, the yield curves of product $B$ do not change as the size of the particles increases.

In effect, it is very simple to verify that when $\phi \to 0$, Equation (13) can be reduced to

$$S_B = \frac{dY_B}{dX_A} = 1 - \left(\frac{mY_B}{1-X_A}\right), \quad (17)$$

with its integration leading to [7]

$$Y_B = \left(\frac{1}{1-m}\right)(1-X_A)\left[(1-X_A)^{(m-1)} - 1\right] \quad (18)$$

Moreover, it is well known for porous catalysts that if $\phi < 0.3$, the effectiveness factor ($\eta$) for the primary reaction does not depend on the particle size [21,37,40].

If $\phi$ is very large, in the region of net intraparticle diffusion control, the effectiveness factor is asymptotically inversely proportional to the particle size ($\eta \to 1/\phi$) [21,37,40]. However, as it can be observed in Figure 1, the yield of product $B$ for a given value of conversion is independent from the particle size, and it can be verified that if $\phi \to \infty$, Equation (13) adopts the following form:

$$S_B = \frac{dY_B}{dX_A} = \frac{1}{\sqrt{m}+1} - \left( \frac{\sqrt{m}\,Y_B}{1-X_A} \right) \tag{19}$$

and, consequently, the yield is [7] as follows:

$$Y_B = \left( \frac{1}{1-m} \right)(1-X_A)\left[ (1-X_A)^{(\sqrt{m}-1)} - 1 \right] \tag{20}$$

For intermediate values of $\phi$, the larger the particle size, the lower the effectiveness factor [21,37,40] and the yield of *B* is negatively affected, as confirmed in Figure 1.

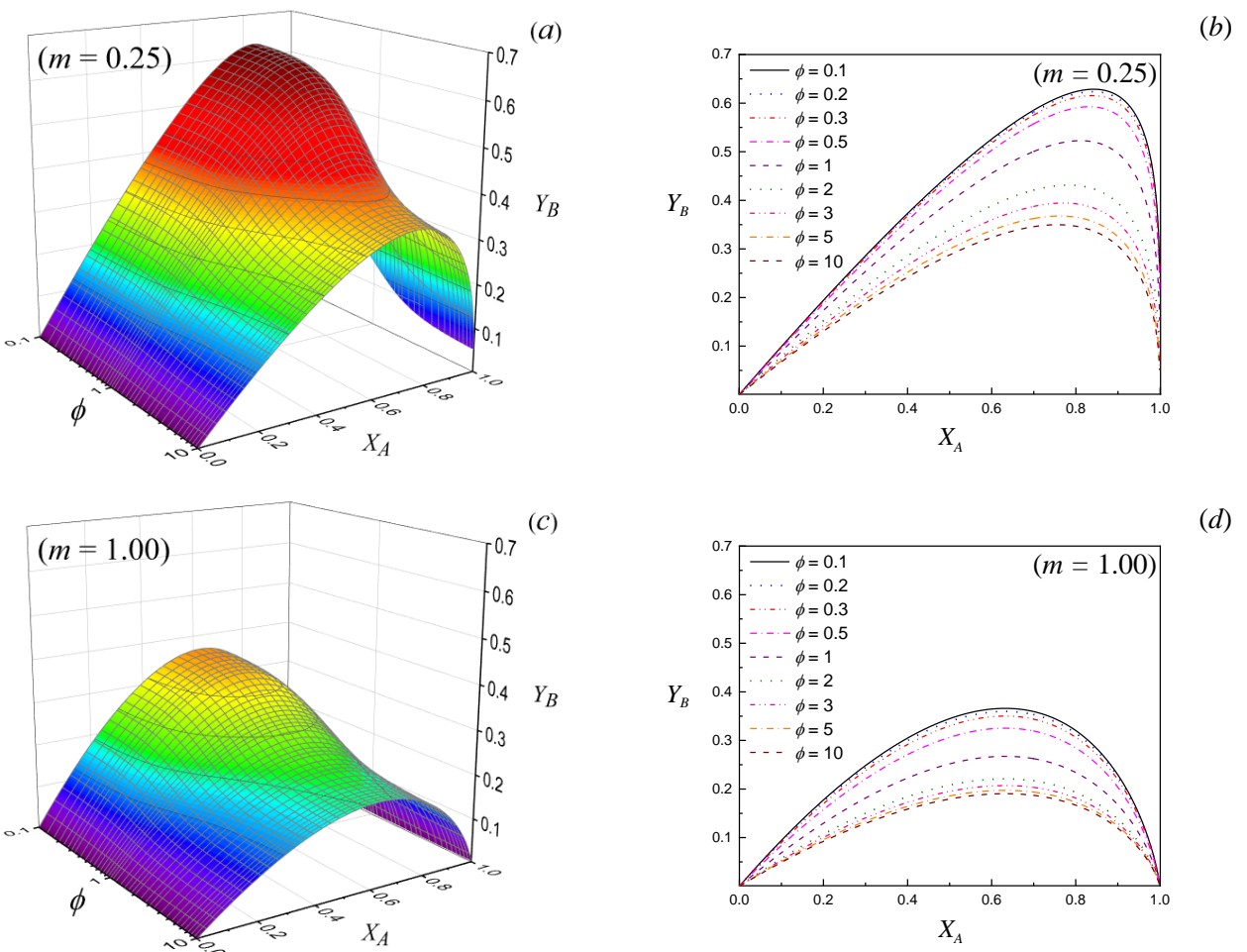

**Figure 1.** Yield of primary product *B* as a function of the conversion of the reactant *A* for different values of the Thiele modulus ($\phi$) and relationships of kinetic constants ($m = k_2/k_1$). (**a**) 3D color map surface $m = 0.25$; (**b**) contour lines for $\phi$ values, $m = 0.25$; (**c**) 3D color map surface $m = 1.00$; (**d**) contour lines for $\phi$ values, $m = 1.00$.

## 3. Non-Uniform Catalyst Particle Sizes: Case of Log-Normal Volume Particle Size Distribution

The mass balances in Section 2.2 were written for a spherical particle with radius *R*. Consequently, the analysis in Section 2.3 is applicable to a catalytic bed of uniform size particles.

Certainly, considering that all the particles in a catalytic bed have exactly the same size is a strong idealization. On the contrary, it has been reported that a great number of catalysts follow a theoretical log-normal PSD [29,32,34,41–44], i.e., the logarithm of the

particle radii is normally distributed [35]. In this case, the volume density function of the particle size (that is, the volume PSD, see Appendix B) is

$$f_{v(R)} = \frac{1}{\sqrt{2\pi}\beta}\left(\frac{1}{R}\right)\exp\left[\frac{-\left(\ln(R)-\alpha\right)^2}{2\beta^2}\right], \qquad \left(0 < R < \infty\right) \qquad (21)$$

As it can be observed from Equation (21), a log-normal distribution is characterized by the following two parameters: the location parameter ($\alpha = \ln(R_{50})$) and the dispersion parameter ($\beta$). $R_{50}$ is the median of the size distribution; that is, 50% volume of the particles in the set have a size larger than $R_{50}$. The parameter $\beta$ represents the standard deviation of the natural logarithm of the particle size, and is indicative of how much the actual distribution deviates from a uniform size (see Figure A1, Appendix B). Table 1 shows the values of the central position ($R_{50}$) and dispersion ($\beta$) parameters that correspond to different cases of commercial catalysts.

While $f_{v(R)}$, given by Equation (21), represents the volume PSD, $f_{v(R)} dR$ is the differential volume fraction of particles with radii between $R$ and ($R + dR$), $f_{(R)}$ is the numerical PSD and $f_{(R)} dR$ is the differential fraction in the number of particles with radii between $R$ and ($R + dR$). As shown in Appendix B (Equations (A12)–(A14)), the volume PSD $f_{v(R)}$ is easily related to the number PSD, $f_{(R)}$ [23,24].

In addition to the hypotheses mentioned in Section 2.1, it will now be assumed that the size distribution of catalyst particles, the particle density and the intrinsic catalytic activity are the same throughout the whole volume of the catalytic bed.

**Table 1.** Parameters of the theoretical log-normal volume PSD in different samples of catalysts calculated from various authors.

| Catalyst | $R_{50}$ (μm) | $\beta$ | Authors | Reference |
|---|---|---|---|---|
| FCC | 26.8–29.6 | 0.199–0.425 | Grace and Sun | [32] |
| FCC | 35.3–54.5 | 0.318–0.613 | Issangya et al. | [29] |
| FCC | 46.3 | 0.429 | Qie et al. | [43] |
| FCC | 27.5 | 0.407 | Rodriguez et al. | [34] |
| IE resin | 284.6 | 0.243 | Dardel | [41] |
| Y zeolite | 0.21–0.30 | 0.134–0.304 | Zhang et al. | [42] |
| Al₂O₃ | 0.50 | 0.678 | Pabst and Gregorová | [44] |

As it was shown in Figure 1, provided a relationship of kinetic constants ($m = k_2/k_1$) is set, the yield of the intermediate product *B* for a given conversion of reactant *A* depends on the Thiele modulus. However, if different sizes ($R$) can be observed in a bed of catalytic particles, it is clear that for a given diffusion-reaction system, a distribution of Thiele moduli $\left(\phi_{(R)}\right)$ will occur. Consequently, if the consecutive reactions represented by Equation (1) take place over a set of particles with a volume PSD $f_{v(R)}$, the contribution to the yield of *B* from those particles with sizes between $R$ and ($R + dR$) is

$$dY_{B(R)} = \left(\int_0^{X_A} S_B \, dX_A\right)_{(R)} f_{v(R)} \, dR \qquad \left(0 < R < \infty\right) \qquad (22)$$

Then, at a given conversion $X_A$, the yield of the intermediate product *B*, resulting from the catalytic particles with different sizes in the bed, is

$$Y_B = \int_0^{\infty}\left(\int_0^{X_A} S_B \, dX_A\right)_{(R)} f_{v(R)} \, dR \qquad (23)$$

For log-normal distributions (see Equation (21)), the following equation can be obtained:

$$Y_B = \int_0^\infty \left( \int_0^{X_A} S_B \, dX_A \right)_{(R)} \left\{ \frac{1}{\sqrt{2\pi}\beta} \left( \frac{1}{R} \right) \exp\left[ \frac{-\left( \ln(R)-\alpha \right)^2}{2\beta^2} \right] \right\} dR \tag{24}$$

Thus, for consecutive reactions with a given relationship between kinetic constants $m = k_2/k_1$, proceeding on a bed of spherical catalytic particles with a log-normal PSD (median $R_{50}$ and dispersion $\beta$), the yield of the primary product $B$ as a function of the conversion of reactant $A$ can be obtained from the following combination of Equations (13) and (24):

$$Y_B = \int_0^\infty \left( \int_0^{X_A} \left( \left\{ \left( \frac{1}{m-1} \right) - \left( \frac{Y_B}{1-X_A} \right) \right\} \left( \frac{\phi_{(R)}\sqrt{m}\coth\left(\phi_{(R)}\sqrt{m}\right)-1}{\phi_{(R)}\coth\left(\phi_{(R)}\right)-1} \right) \right. \right.$$
$$\left. \left. -\left( \frac{1}{m-1} \right) \right) dX_A \right) \left\{ \frac{1}{\sqrt{2\pi}\beta} \left( \frac{1}{R} \right) \exp\left[ \frac{-\left( \ln(R)-\alpha \right)^2}{2\beta^2} \right] \right\} dR \tag{25}$$

It is important to note that the dependency on the radius of the catalyst particles in the integrand of Equation (25) is not only due to the PSD (Equation (21)), but also to the selectivity itself (see Equation (13)), which involves the Thiele modulus $\left( \phi_{(R)} \right)$, in turn depending on the particle size. As described, the characteristic length in the generalized Thiele modulus (Equation (8)) is expressed by the relationship between the volume and the area of a sphere with radius $R$ [37], which is as follows:

$$L_{(R)} = \frac{V_{p(R)}}{A_{p(R)}} = \frac{\left( \frac{4}{3}\pi R^3 \right)}{\left( 4\pi R^2 \right)} = \frac{R}{3} \tag{26}$$

and then

$$\phi_{(R)} = \frac{R}{3}\sqrt{\frac{k_1}{D}} \tag{27}$$

The average volume and area of a set of particles with different sizes are

$$V_p^{(PSD)} = \int_0^\infty \left( \frac{4}{3}\pi R^3 \right) f_{(R)} \, dR = \frac{4}{3}\pi \overline{R^3} \tag{28}$$

and

$$A_p^{(PSD)} = \int_0^\infty \left( 4\pi R^2 \right) f_{(R)} \, dR = 4\pi \overline{R^2} \tag{29}$$

where the third and second order momenta of the PSD (Equation (A13), Appendix B) can be recognized in Equations (28) and (29), respectively.

Analogously to Equation (26), it is possible to adopt a characteristic length for the set of particles with a certain PSD as the relationship between the average volume (Equation (28)) and area (Equation (29)) of the particles [24].

$$L^{(PSD)} = \frac{V_p^{(PSD)}}{A_p^{(PSD)}} = \frac{1}{3}\frac{\overline{R^3}}{\overline{R^2}} \tag{30}$$

By inspection of Equation (A16) (Appendix B), it is easy to recognize in Equation (30) that

$$L^{(PSD)} = \frac{V_p^{(PSD)}}{A_p^{(PSD)}} = \frac{R_m}{3} \tag{31}$$

where $R_m$ is the mean harmonic radius, that is, the radius of spherical particles in a bed with uniform particle size, whose relationship $V_{p(R_m)}/A_{p(R_m)}$ is the same as that of a bed

of particles with a size distribution of $V_p^{(\mathrm{PSD})}/A_p^{(\mathrm{PSD})}$ [23,24]. Then, for particles with a certain number $\left(f_{(R)}\right)$ or volume $\left(f_{\mathrm{v}(R)}\right)$ PSD, a representative value of the Thiele modulus of the primary reaction ($A \xrightarrow{k_1} B$) can be expressed in terms of this characteristic length, in a similar way to Equation (8).

$$\phi_m = \frac{V_p^{(\mathrm{PSD})}}{A_p^{(\mathrm{PSD})}} \sqrt{\frac{k_1}{D}} = \frac{R_m}{3} \sqrt{\frac{k_1}{D}} \tag{32}$$

Using $\phi_m$ is advantageous, as shown by Haynes [23] for first-order kinetics and García et al. [24] for more complex kinetics, including the power-law and Langmuir-Hinshelwood-Hougen-Watson type. It has been verified that even though the plots of the catalytic effectiveness factor ($\eta$) as a function of $\phi_m$ differ for each value of the dispersion ($\beta$) in the PSD, all the curves have the same asymptotic behavior for the extreme situations of chemical control ($\eta \to 1$ when $\phi_m << 0.1$) or intraparticle diffusion control ($\eta \propto 1/\phi_m$ when $\phi_m >> 3$), including, of course, the particular case of uniform particle size, that is, $\beta \to 0$ [23,24]. Then, for the subsequent analysis, it is convenient to express the size of the spherical particles (*R*) as a function of a standard log-normal variable (*z*, see Equation (A19), Appendix B), with the mean harmonic radius (*R_m*) and the standard deviation of the natural logarithm of the particle size (*β*) as parameters, that is, $R = R_m \exp\left(\sqrt{2}\beta z + \beta^2/2\right)$ (Equation (A22), Appendix B). Similarly, the Thiele modulus for a given particle size $\phi_{(R)}$ can be expressed in terms of the Thiele modulus with a mean harmonic radius of its characteristic length, i.e., $\phi_{(z)} = \phi_m \exp\left(\sqrt{2}\beta z + \beta^2/2\right)$ (Equation (A24), Appendix B). BY introducing this change in variables, Equation (25) can be equivalently written as

$$Y_B = \int_{-\infty}^{\infty} \left( \int_0^{X_A} \left( \left\{ \left(\frac{1}{m-1}\right) - \left(\frac{Y_B}{1-X_A}\right) \right\} \right. \right.$$
$$\left. \left. \left( \frac{\phi_{(\phi_m, z)}\sqrt{m}\coth\left(\phi_{(\phi_m, z)}\sqrt{m}\right) - 1}{\phi_{(\phi_m, z)}\coth\left(\phi_{(\phi_m, z)}\right) - 1} \right) - \left(\frac{1}{m-1}\right) \right) dX_A \right) \left\{ \frac{\exp\left(-z^2\right)}{\sqrt{\pi}} \right\} dz \tag{33}$$

The following tree parameters occur in Equation (33): the Thiele modulus for the primary reaction, based on a characteristic length that represents the whole set of particle sizes in the bed $\left(\phi_m\right)$, the dispersion of those particle sizes $\left(\beta\right)$ and the relationship between the kinetic constants of the secondary and primary reactions $\left(m\right)$.

## 4. Results and Discussion

Figures 2 and 3 show how the yield of the primary product *B* changes as a function of the conversion of the reactant *A* for different values of the parameters $\phi_m$, $\beta$ and *m*. A number of similarities to the case of uniform particle size (see Figure 1) can be observed. In effect, the following conclusions can be drawn:

- In all the cases, as expected, initially $Y_B$ increases as $X_A$ increases, reaching a maximum value and then decreasing at higher conversions.
- If a relationship $m = k_2/k_1$ is set, at a certain conversion ($X_A$), the larger the Thiele modulus, the lower the yield of *B*.
- For a known pair of parameters, $\phi_m$ and $\beta$, the higher the relationship $m = k_2/k_1$, the lower the maximum yield of product *B*, which is achieved at lower conversions.

It can be observed that if the situation of uniform particle size is compared against the cases where a distribution of sizes exists, for a given pair of values of $\phi_m$ and $m$, the yield curves are always higher in the case of a bed with uniform particle size, that is, $\beta = 0$. Moreover, the more significant the dispersion, the lower the yield of $B$ at a given conversion.

It was previously shown for simple reactions $A \rightarrow P$ that occur on catalysts with log-normal volume PSD that the values of the effectiveness factor $\eta$, as a function of the Thiele modulus $\phi_m$, besides being asymptotic at very small (chemical control) and very large (diffusion control) $\phi_m$ values, are always smaller than those of the uniform size case [23,24]. Moreover, the larger the dispersion, the smaller the effectiveness factor. This observation can be explained if it is considered that in log-normal volume PSD, $R_m < R_{50}$ always (see Equation (A21), Appendix B). As the radius of half the mass or volume of the bed particles is larger than $R_{50}$, the proportion of particles with radii larger than $R_m$ is higher than 50%. For example, it can be observed from Figure A1 (Appendix B) that the mass or volume percentages of particles with radii larger than $R_m$ are 59.9% ($\beta = 0.50$), 64.6% ($\beta = 0.75$) and 69.1% ($\beta = 1.00$). Therefore, the resulting global effectiveness of a bed with particles that show a certain dispersion in their PSD and a harmonic mean radius $R_m$ is always lower than that of the bed that has particles with uniform sizes that are the same as $R_m$; moreover, the more important the dispersion, the more significant the difference.

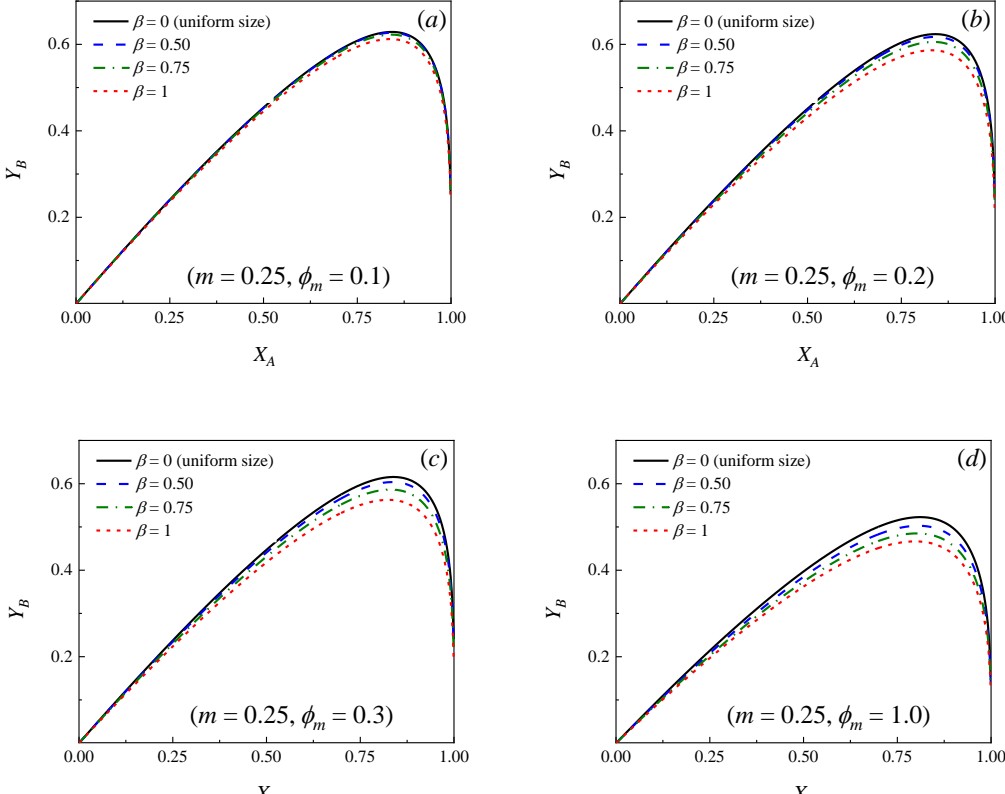

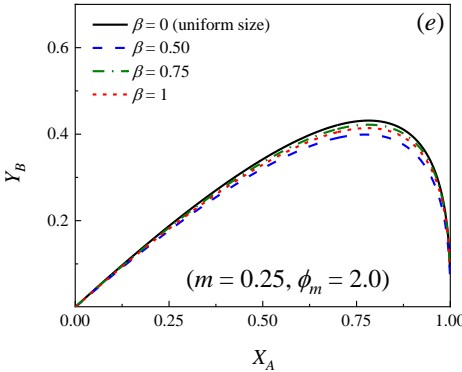
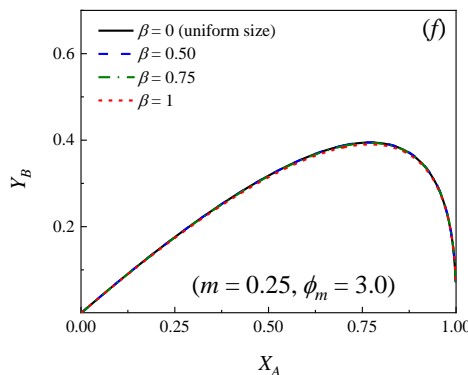

**Figure 2.** Yield of primary product *B* as a function of the conversion of the reactant *A* for different values of the dispersion parameter ($\beta$). $m = k_2/k_1 = 0.25$. (**a**) $\phi_m = 0.1$; (**b**) $\phi_m = 0.2$; (**c**) $\phi_m = 0.3$; (**d**) $\phi_m = 1.0$; (**e**) $\phi_m = 2.0$; (**f**) $\phi_m = 3.0$.

This view can also be applied to series reactions to explain the lower yields of intermediate product *B* due to the effect of PSD, as shown in Figures 2 and 3. In effect, given that the more dispersed the PSD, the higher the proportion of particles larger than $R_m$ (the length that characterizes the Thiele modulus, $\phi_m$) (Figure A1, Appendix B), the diffusion limitations both on reactant *A* and product *B* will be more important. It was observed in Figure 1, for particles with increasing uniform size, that the more important diffusion restrictions negatively impact the yield of the intermediate product *B* [7].

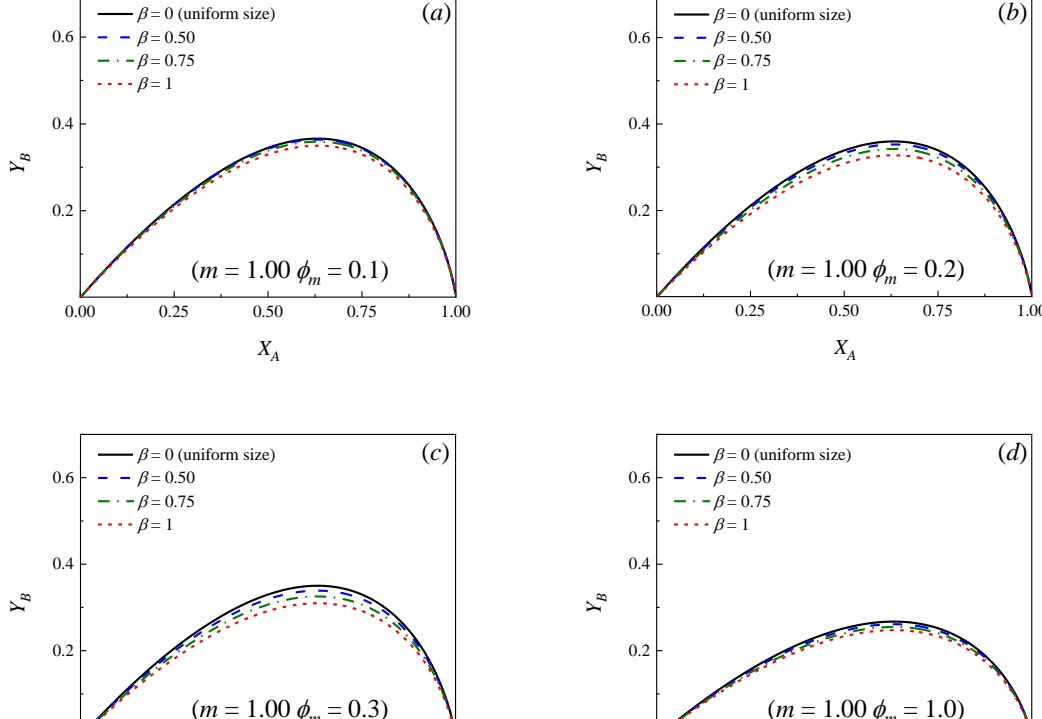

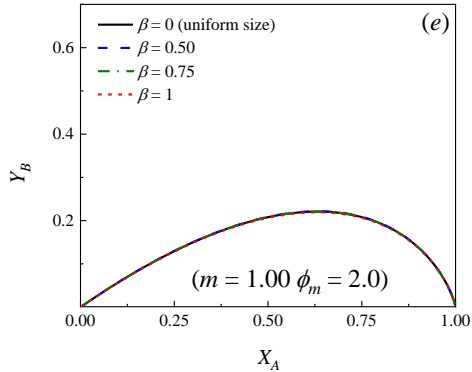
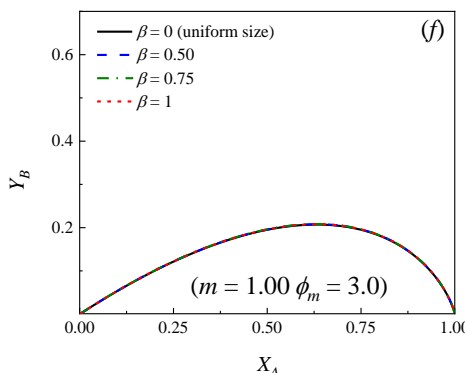

**Figure 3.** Yield of primary product *B* as a function of the conversion of the reactant *A* for different values of the dispersion parameter ($\beta$). $m = k_2/k_1 = 1.00$. (**a**) $\phi_m = 0.1$; (**b**) $\phi_m = 0.2$; (**c**) $\phi_m = 0.3$; (**d**) $\phi_m = 1.0$; (**e**) $\phi_m = 2.0$; (**f**) $\phi_m = 3.0$.

Figures 2 and 3 show that, independently from the relationship $m = k_2/k_1$, the dispersion in the PSD does not influence significantly the yields when $\phi_m$ is larger than approximately 3. This means that, even though the particles in the bed have many different sizes, essentially, all of them exert high diffusion resistances, both for the reactant and the primary product and the yield of *B* can be calculated directly from Equation (16) [7,9], ignoring the dispersion, provided $\phi_m > 3$. Moreover, it can be observed in Figures 2 and 3 that when $\phi_m < 0.1$ (chemical control), the yield curves tend to overlap and the yield of *B* can be calculated directly from Equation (18), independently from the value of $\beta$.

The case where *m* is much smaller than about 0.25 is close to the case of a simple reaction *A→B*. Under this scenario, the extent of the secondary reaction is essentially negligible over a wide range of conversions of reactant *A* (see Figure S1 in Supplementary Materials, $m = 0.10$); therefore, it no longer makes sense to analyze the system under the scheme of consecutive reactions. In contrast, a system of consecutive reactions, where the relationship *m* is much larger than about 1, is surely not attractive, as yields of the intermediate product are very low (see Figure S2 in Supplementary Materials, $m = 2.00$).

## 5. Conclusions

In a similar manner to catalytic beds with uniform size particles, beds with particles of different sizes show that the yield of the intermediate product *B* in a system of consecutive reactions *A→B→C*, where both the reactant and products are subjected to diffusion limitations, reaches a maximum value as a function of the conversion of reactant *A*. The yield then decreases as a consequence of the prevalence of the secondary reaction to product *C*.

If the reaction system includes first-order irreversible reactions that proceed on spherical catalyst particles with a log-normal volume particle size distribution, which are common in many catalytic applications, the following three parameters characterize the system: $\phi_m$, *m* and $\beta$. $\phi_m = (R_m/3)\sqrt{k_1/D}$ is the Thiele modulus for the primary reaction, where $R_m$ is the mean harmonic radius of the PSD. *m* is the relationship between the intrinsic kinetic constants for the secondary and primary reaction and $\beta$ represents the standard deviation of the natural logarithm of the particle size, indicative of the dispersion of sizes in the distribution. The selection of the mean harmonic radius as the characteristic length that is representative of the whole set of particles in the calculation of the Thiele modulus is completely adequate.

If intraparticle diffusion resistances affect the chemical species, the selectivity to the intermediate product *B* is negatively influenced by the dispersion in PSD, given *m* and $\phi_m$. The larger the dispersion ($\beta$) in the PSD, the stronger the negative impact. However, in

extreme situations, such as the absence of diffusion restrictions (small $\phi_m$) or under net intraparticle diffusion control (large $\phi_m$), the dispersion in the particle size ($\beta$) does not affect the yield curves of the intermediate product *B*.

**Supplementary Materials:** The following supporting information can be downloaded at: https://www.mdpi.com/article/10.3390/catal12101214/s1, Figure S1: Yield of primary product *B* as a function of the conversion of the reactant *A* for different values of the dispersion parameter ($\beta$). $m = k_2/k_1 = 0.10$. (a) $\phi_m = 0.1$; (b) $\phi_m = 0.2$; (c) $\phi_m = 0.3$; (d) $\phi_m = 1.0$; (e) $\phi_m = 2.0$; (f) $\phi_m = 3.0$; Figure S2: Yield of primary product *B* as a function of the conversion of the reactant *A* for different values of the dispersion parameter ($\beta$). $m = k_2/k_1 = 2.00$. (a) $\phi_m = 0.1$; (b) $\phi_m = 0.2$; (c) $\phi_m = 0.3$; (d) $\phi_m = 1.0$; (e) $\phi_m = 2.0$; (f) $\phi_m = 3.0$.

**Author Contributions:** Conceptualization, J.R.G., C.M.B. and U.S.; software, J.R.G.; validation, J.R.G. and C.M.B.; formal analysis, J.R.G., C.M.B. and U.S.; resources, J.R.G. and U.S.; writing—original draft preparation, J.R.G.; writing—review and editing, J.R.G., C.M.B. and U.S.; visualization, J.R.G., C.M.B. and U.S.; supervision, U.S.; project administration, U.S.; funding acquisition, J.R.G. and U.S. All authors have read and agreed to the published version of the manuscript.

**Funding:** This work has been carried out with financial support from the Universidad Nacional del Litoral (UNL, Santa Fe, Argentina), Proj. CAID 50620190100177LI; Consejo Nacional de Investigaciones Científicas y Técnicas (CONICET), Proj. PIP 11220200103146CO; and Agencia Nacional de Promoción de la Investigación, el Desarrollo Tecnológico y la Innovación (Agencia I+D+i), Proj. PICT 3391/2019 and Proj. PICT 2074/2020.

**Data Availability Statement:** Not applicable.

**Conflicts of Interest:** The authors declare no conflict of interest.

## Abbreviations

*Symbols*

| | |
|---|---|
| $A$ | area (m²) |
| $C$ | concentration (gmol/m³) |
| $D$ | effective diffusion coefficient (m²/s) |
| $f$ | particle size distribution (1/m) |
| $F$ | cumulative particle size distribution (dimensionless) |
| $k$ | overall kinetic constant (1/s) |
| $L$ | characteristic length (m) |
| $m$ | relationship between the intrinsic kinetic constants (dimensionless) |
| $P$ | generic product |
| $R$ | catalyst particle radius (m) |
| $r$ | radial distance (m) |
| $S$ | selectivity (dimensionless) |
| $V$ | volume (m³) |
| $X$ | conversion (dimensionless) |
| $Y$ | yield (dimensionless) |
| $z$ | standard log-normal variable (dimensionless) |

*Greek symbols*

| | |
|---|---|
| $\alpha$ | location parameter in log-normal volume particle size distribution |
| $\beta$ | dispersion parameter in log-normal volume particle size distribution |
| $\chi$ | dimensionless concentration in the fluid phase |
| $\phi$ | Thiele modulus |
| $\eta$ | effectiveness factor |
| $\rho$ | dimensionless radial distance |
| $\xi$ | dimensionless concentration in the catalyst particle |

*Subscripts*

| | |
|---|---|
| 1 | refers to primary reaction |
| 2 | refers to secondary reaction |

| 50 | refers to the median of a particle size distribution |
|---|---|
| *A* | refers to reactant |
| *B* | refers to primary product |
| *f* | fluid phase |
| *m* | refers to the mean harmonic value |
| *p* | particle |
| *R* | refers to a specific particle size |
| v | volume |
| *Superscripts* | |
| ° | inlet or initial concentration |
| – | refers to the *x*-th order momentum of a continuous distribution |
| PSD | refers to the particle size distribution |
| *x* | *x*-th order momentum |

**Appendix A**

By using the following definitions of dimensionless variables,

$$\rho = \frac{r}{\left(V_p / A_p\right)} = \frac{3\,r}{R} \tag{A1}$$

$$\xi_A = \frac{C_{A(r)}}{C_A^\circ}, \qquad \xi_B = \frac{C_{B(r)}}{C_A^\circ} \tag{A2}$$

$$\chi_A = \frac{C_{A\,f}}{C_A^\circ}, \qquad \chi_B = \frac{C_{B\,f}}{C_A^\circ} \tag{A3}$$

the mass balances in the catalyst particles for reactant *A* (Equation (2)) and primary product *B* (Equation (5)), and their corresponding boundary conditions, can be written dimensionless as

$$\frac{1}{\rho^2} \frac{d}{d\rho}\left(\rho^2 \frac{d\xi_A}{d\rho}\right) = \phi_1^{\,2} \xi_A \qquad (\,0 < \rho < 3\,) \tag{A4}$$

$$\frac{d\xi_A}{d\rho} = 0 \qquad (\,\rho = 0\,) \tag{A5}$$

$$\xi_A = \chi_A \qquad (\,\rho = 3\,) \tag{A6}$$

$$\frac{1}{\rho^2} \frac{d}{d\rho}\left(\rho^2 \frac{d\xi_B}{d\rho}\right) = -\phi_1^{\,2}\, \xi_A + \left(\phi_1^{\,2} m\right)\xi_B \qquad (\,0 < \rho < 3\,) \tag{A7}$$

$$\frac{d\xi_B}{d\rho} = 0 \qquad (\,\rho = 0\,) \tag{A8}$$

$$\xi_B = \chi_B \qquad (\,\rho = 3\,) \tag{A9}$$

where

$$\phi_{1(R)} = \frac{V_{p(R)}}{A_{p(R)}} \sqrt{\frac{k_1}{D}} \tag{A10}$$

is the Thiele modulus for the primary reaction ($A \xrightarrow{\ k_1\ } B$), and

$$m = \frac{k_2}{k_1} \tag{A11}$$

is the relationship between the intrinsic kinetic constants for the secondary $(B \to C)$ and primary $(A \to B)$ reactions. It is important to note that, in the right hand side of Equation (A7), the term that expresses the consumption of product $B$ by the chemical reaction shows the coefficient $\phi_1^2 m = \left( L^2 \dfrac{k_1}{D} \right) \left( \dfrac{k_2}{k_1} \right) = \phi_2^2$, $\phi_2$ being the Thiele modulus for the secondary reaction ($B \xrightarrow{k_2} C$). Then, $m = \sqrt{\phi_2/\phi_1}$.

For the sake of clarity, the subscript 1 in the Thiele modulus for the primary reaction is omitted in the text, i.e., $\phi_{1(R)} = \phi_{(R)}$.

**Appendix B**

The meaning of the volume distribution of particle sizes ($f_{v(R)}$, Equation (21)) can be understood, considering that $f_{v(R)} \, dR$ is the differential volume fraction of particles with radii between $R$ and $(R + dR)$. That is,

$$f_{v(R)} \, dR = \frac{\left( \dfrac{4}{3} \pi R^3 \right) f_{(R)} \, dR}{\displaystyle\int_0^\infty \left( \dfrac{4}{3} \pi R^3 \right) f_{(R)} \, dR} = \frac{R^3 f_{(R)} \, dR}{\displaystyle\int_0^\infty R^3 f_{(R)} \, dR} \tag{A12}$$

Instead, $f_{(R)}$ is the number distribution of particle sizes, implying that $f_{(R)} \, dR$ is the differential fraction in the number of particles with radii between $R$ and $(R + dR)$.

The $x$-th order momentum of a continuous distribution $f_{(R)} \, dR$ [35] is given by

$$\overline{R^x} = \int_0^\infty R^x f_{(R)} \, dR \tag{A13}$$

Then, it is clear from Equation (A12) that the relationship between the volume and the number distributions of particle sizes is

$$f_{v(R)} = \frac{R^3}{\overline{R^3}} f_{(R)} \tag{A14}$$

where, for a given PSD, the third-order momentum is constant [35].

By using Equation (A14) in Equation (A13), the second-order momentum ($x = 2$) is

$$\overline{R^2} = \int_0^\infty R^2 f_{(R)} \, dR = \overline{R^3} \int_0^\infty \left( \frac{1}{R} \right) f_{v(R)} \, dR \tag{A15}$$

It was shown in Equation (28) that the average volume of a set of particles with a number distribution of particle sizes $f_{(R)}$ is proportional to $\overline{R^3}$, while the average area is proportional to $\overline{R^2}$ (Equation (29)). Then, according to Equation (30), it can be observed that the characteristic length in the generalized Thiele modulus includes the relationship $\overline{R^3}/\overline{R^2}$, which can be visualized from Equation (A15) to be

$$R_m = \frac{\overline{R^3}}{\overline{R^2}} = \frac{1}{\displaystyle\int_0^\infty (1/R) f_{v(R)} \, dR} \tag{A16}$$

$R_m$ is the mean harmonic radius, which can be easily calculated from Equation (A16) if the $f_{v(R)}$ or $f_{(R)}$ PSD are known.

Different authors [29,32,34,41–44] reported that the particle sizes in different catalysts follow log-normal volume distributions.

$$f_{v(R)} = \frac{1}{\sqrt{2\pi}\beta}\left(\frac{1}{R}\right)\exp\left[\frac{-\left(\ln(R)-\alpha\right)^2}{2\beta^2}\right] \qquad \left(0 < R < \infty\right) \qquad (A17)$$

where $\alpha$ is the natural logarithm of the median ($R_{50}$) and $\beta$ is a dispersion parameter that quantifies how much the actual distribution diverts from a uniform size.

The cumulative distribution function of a log-normal volume distribution $\left(F_{v(R)}\right)$ represents the volume fraction in the bed of those particles with sizes smaller than or equal to $R$. From the definition of the error function [45], it can be shown that

$$F_{v(R)} = \int_0^R f_{v(R)}\,dR = \frac{1}{2} + \frac{1}{2}\,\mathrm{erf}\left[\frac{\ln(R)-\alpha}{\sqrt{2}\beta}\right] \qquad \left(0 < R < \infty\right) \qquad (A18)$$

Figure A1 shows the plots of differential (Equation (A17)) and cumulative (Equation (A18)) log-normal volume particle size distributions with different dispersions ($\beta$).

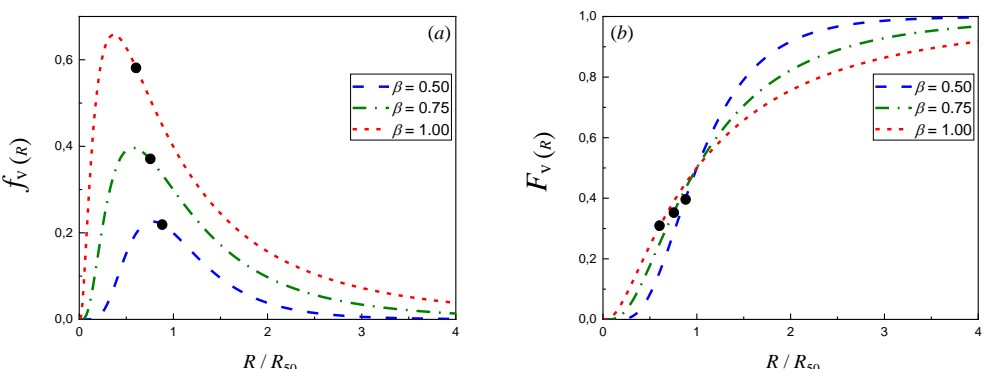

**Figure A1.** Differential $f_{v(R)}$ (**a**) and cumulative $F_{v(R)}$ (**b**) log-normal volume PSD with different dispersions ($\beta$). The thick dots indicate the location of $R_m/R_{50}$ (Equation (A21)).

In practice, it is helpful to use the standard log-normal variable ($z$) in order to standardize the actual (dimensional) particle size ($R$) [23,35].

$$z = \frac{\ln(R)-\alpha}{\sqrt{2}\beta} = \frac{\ln\left(R/R_{50}\right)}{\sqrt{2}\beta} \qquad (A19)$$

According to this variable transformation, the particle size ($R$) can be written in terms of $z$, with the median ($R_{50}$) and the standard deviation of the natural logarithm of the sizes ($\beta$) as parameters.

$$R = \exp\left(\alpha + \sqrt{2}\beta z\right) = R_{50}\exp\left(\sqrt{2}\beta z\right) \qquad (A20)$$

By replacing $R$ from Equation (A20) and $f_{v(R)}$ from Equation (A17) in Equation (A16), it is possible to obtain an expression for the mean harmonic radius of a log-normal volume distribution [23], which yields

$$R_m = \exp\left(\alpha - \beta^2/2\right) = R_{50}\exp\left(-\beta^2/2\right) \qquad (A21)$$

and, by combining this expression with Equation (A20), one can obtain the following equation:

$$R = R_m \exp\left(\sqrt{2}\beta z + \beta^2/2\right) \qquad (A22)$$

If the particle radius $R$ given by Equation (A22) is introduced into Equation (27), the generalized Thiele modulus is

$$\phi_{(z)} = \frac{R_m}{3}\sqrt{\frac{k_1}{D_p}}\exp\left(\sqrt{2}\beta z + \frac{\beta^2}{2}\right) \tag{A23}$$

and, according to Equation (32), this last expression can be rewritten into

$$\phi_{(\phi_m, z)} = \phi_m \exp\left(\sqrt{2}\beta z + \frac{\beta^2}{2}\right) \tag{A24}$$

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
