# Peer review of "Impact of the Non-Uniform Catalyst Particle Size on Product Selectivities in Consecutive Reactions"

_catalysts, doi:10.3390/catal12101214_

Round 1

Reviewer 1 Report

The paper deals with the investigation about how the particle size distribution can affect selectivity in consecutive reactions. The paper is formally well written and the mathematics is rigorous. The results are interesting, and they could be useful for future applications. Few concerns I have on the paper that should be addressed prior publication:

1.      The range of the investigated parameters. The model gives responses that show only minor effect on the main parameters investigated. I suggest to re-run the simulations widening the ranges of applications, to show the real potentialities of the model.

2.      The system description section includes some trivial information that could be removed.

Reviewer 2 Report

In this manuscript (catalysts-1932250), the authors studied the catalyst particle size on the product selectivity for the consecutive reaction. It is a very interesting and important topic in heterogeneous catalysis.  The data are systematically displayed and explained. I can recommend the publication on this journal, but there are several small comments needed to be tackled before publication.

1. for a modeling paper, the error bars, confidence intervals should be included and reported, then people can have a better overview of the plot and data. And this is my only big concern about this manuscript.

2. for the figures, it is better if the authors can add the annotation to each figure.

Round 2

Reviewer 1 Report

The authors addressed the reviewer comments. It is now publishable.